**Data Availability Statement:** Data supporting the findings of this study have not been made generally

# What support do frontline workers want? A qualitative study of health and social care workers' experiences and views of psychosocial support during the COVID-19 pandemic

Jo Billings[1]*, Nada Abou Seif[1], Siobhan Hegarty[1], Tamara Ondruskova[1], Emilia Soulios[1], Michael Bloomfield[1,2,3,4], Talya Greene[1,5]

1 Division of Psychiatry, University College London, London, United Kingdom, 2 Traumatic Stress Clinic, Camden & Islington NHS Foundation Trust, London, United Kingdom, 3 National Institute for Health Research University College London Hospitals Biomedical Research Centre, London, United Kingdom, 4 University College London Hospitals NHS Foundation Trust, London, United Kingdom, 5 Department of Community Mental Health, University of Haifa, Haifa, Israel

* j.billings@ucl.ac.uk

## Abstract

### Background

The COVID-19 pandemic has placed a significant burden on the mental health and wellbeing of frontline health and social care workers. The need to support frontline staff has been recognised. However, there is to date little research specifically on how best to support the mental health needs of frontline workers, and none on their own experiences and views about what might be most helpful.

### Aims

We set out to redress this research gap by qualitatively exploring UK frontline health and social care workers' own experiences and views of psychosocial support during the pandemic.

### Method

Frontline health and social care workers were recruited purposively through social media and by snowball sampling via healthcare colleagues. Workers who volunteered to take part in the study were interviewed remotely following a semi-structured interview guide. Transcripts of the interviews were analysed by the research team following the principles of Reflexive Thematic Analysis.

### Results

We conducted 25 interviews with frontline workers from a variety of professional groups working in health and social care settings across the UK. Themes derived from our analysis

available due to the personal and sensitive content of the frontline workers' accounts, in line with the requirements stipulated by the UCL Research Ethics Committee. Data can be made available upon reasonable request to the corresponding author (JB) or to the UCL Research Ethics Committee: ethics@ucl.ac.uk, 2 Taviton St, London WC1E 6BT.

**Funding:** The authors received no specific funding for this work.

**Competing interests:** The authors have declared that no competing interests exist.

showed that workers' experiences and views about psychosocial support were complex. Peer support was many workers' first line of support but could also be experienced as a burden. Workers were ambivalent about support shown by organisations, media and the public. Whilst workers valued psychological support services, there were many disparities in provision and barriers to access.

## Conclusions

The results of this study show that frontline health and social care workers are likely to need a flexible system of support including peer, organisational and professional support. More research is needed to fully unpack the structural, systemic and individual barriers to accessing psychosocial support. Greater collaboration, consultation and co-production of support services and their evaluation is warranted.

## 1. Introduction

On March 11[th], 2020, the World Health Organisation declared severe acute respiratory syndrome coronavirus 2 (SARS-CoV-2), the virus causing COVID-19, to be a worldwide pandemic [1]. In the UK, healthcare workers braced themselves, as news from around the world described healthcare workers overcome by the requirement to treat rapidly growing numbers of patients affected by the virus. Many NHS services in the UK did subsequently report themselves to be overwhelmed, whilst other services mobilised to respond to additional demand never reached expected capacity. Other groups, such as care home staff, were overwhelmed in ways never anticipated.

Early quantitative research emerging from the UK [2–4] and around the world [5, 6] has demonstrated a significant mental health burden experienced by frontline workers in response to COVID-19, with elevated rates of depression, anxiety, post-traumatic stress disorder (PTSD) and suicidality reported. In the UK, it has been estimated that 45–58% of the frontline health and social care workforce met criteria for clinically significant levels of anxiety, depression and/or PTSD shortly following the first wave of the pandemic [2, 3]. This is amongst a workforce already under considerable strain pre-COVID-19, as evidenced by the growing incidence of stress, burnout, depression, drug and alcohol dependence and suicide across all groups of health professionals, worldwide [7].

The need to support the mental health of frontline staff during COVID-19 has been recognised [8]. However, this pandemic has also highlighted a paucity of research on the mental health needs of frontline health and social care workers, and a lack of evidence-based guidance about what psychosocial support might be most effective in helping them. What guidance there has been issued during the COVID-19 crisis has tended to come from expert opinion and be extrapolated from other professional groups, such as military personnel. In a rapidly escalating pandemic, such top down 'command and control' guidance has been of value. However, health and social care workers are a unique workforce, living alongside their work, not deployed into time-limited situations of crisis, nor with any allocated time to decompress [9]. We require specific evidence on how best to support the mental health needs of this population, during a pandemic and beyond, and this evidence-base must necessarily be underpinned by the workforce's own views and preferences.

So far, there has been very little qualitative research exploring frontline workers own experiences and views of working on COVID-19. What qualitative research has been published so

far has tended to be on small samples, limited to doctors and nurses, and of poor to moderate quality [10]. No research to date has considered health and social care workers' own opinions about psychosocial support. We set out to redress this gap by asking frontline health and social care workers about their own experiences and views about psychosocial support during the pandemic, and in so doing, aimed to amplify the voices and views of frontline workers themselves. This collaborative contribution is vital as we face further waves of COVID-19 in the UK and need to ensure that we can maintain the wellbeing of the health and social care workforce and sustain the services they provide to the UK population.

## 2. Method

### 2.1 Study design, participants, and procedures

All procedures were approved by the University College London Research Ethics Committee (Ref. 18341/001).

We recruited frontline health and social care workers purposively through social media (Twitter and Facebook) and by snowball sampling via healthcare colleagues, asking them to pass on details of the study to potential participants. We invited any health and social staff who had been working directly to treat patients affected by COVID in any UK based health or social care organisation. We excluded health and social care workers not working in direct COVID related roles or other non-health or social care frontline workers. We deliberately sought a wide range of participants, including different professional groups, career stages and geographical locations, to access a diverse range of experiences and views.

Potential participants were invited to contact the first author via email. They were then sent the study Participant Information Sheet and Consent Form by return email. As all interviews took place remotely, participants were asked to return the electronically signed consent form to the allocated interviewer in advance of the interview. All interviews were completed by four graduate students in Clinical Mental Health who received training from the first author. Written informed consent was obtained from all participants prior to taking part in the interviews.

Interviews were arranged at a mutually convenient time between the interviewer and the frontline worker and took place by telephone or online video call. The interview guide (see S1 Data) was drafted collaboratively by the research team, in consultation with our Expert Reference Group, comprising NHS service leads, Wellbeing officers, and clinicians with lived experience of mental health difficulties. All the interviews were audio recorded and then transcribed verbatim by the interviewer who conducted the interview. All potentially identifying information about the participant and their place of work was removed from the interview transcripts to protect participants' anonymity. Pseudonyms are used for illustrative quotes.

The wellbeing of our participants and our research team was of paramount importance throughout this study. If any participant reported significant distress, they were signposted to local and National sources of psychological support. The research team were supported by regular supervision from the first author.

### 2.2 Analysis

We followed the principles of reflexive thematic analysis [11, 12], underpinned by a critical realist epistemological stance, throughout this study. In keeping with thematic analysis, we sought immersion in the data by reading and re-reading all the transcripts, reflecting on the interviews and discussing emerging themes in research team meetings. JB and the four interviewers independently coded two transcripts each, to derive an initial list of potential codes. This coding frame was reviewed and agreed through discussion in the team. All transcripts were then imported into NVivo Pro V12 and coded into the provisional coding frame, which

was further extended and refined with the coding of subsequent transcripts. All coding was inductive, derived from the data, and not pre-determined by any pre-existing theories. A final set of themes was then developed from the coded data and revised with feedback from participants, health and social care worker colleagues and our Expert Reference Group.

The concept of 'data saturation' is not considered consistent with the values and assumptions of Reflexive Thematic Analysis and its underlying non-Positivist ontology [13]. Therefore, we drew on an 'information power' approach [14], considering the research aims, sample specificity, established theory, quality of dialogue and analysis strategy to guide our decisions about how many participants to interview. Ultimately, we sought a sufficient sample size of health and social care workers to offer a pragmatic balance between breadth, and depth, of analysis, as advocated by Braun and Clarke [13] the founders of Reflexive Thematic Analysis.

## 2.3 Reflexivity

Reflexivity is an important component of all qualitative research, enabling the reader to consider the validity of the analysis by better understanding the research team who have produced it. This team is made up of a diverse group of researchers, including different career stages, clinical specialities, genders, and cultural groups. JB is a Consultant Clinical Psychologist and Associate Clinical Professor with over 20 years of experience of working in the NHS. She has particular expertise in the mental health and wellbeing of high-risk occupational groups. NAS, SH, TO and ES are all MSc graduates in Clinical Mental Health Sciences. All volunteered to work with the team during the COVID-19 pandemic. It is worth noting that all the interviewers who conducted the interviews are female. MB is an Honorary Consultant Psychiatrist and Principal Clinical Research Fellow. He has 19 years' experience of working clinically in the NHS. TG is a Senior Lecturer specialising in research on psychological responses to mass traumatic events. The research team was an ethnically diverse group, including White British, Arab, White Irish and White European backgrounds. As a team we brought a range of different perspectives and experiences to this topic.

## 2.4 Quality and validity

We have adhered to the highest standards for conducting and writing up qualitative research throughout this study. We have drawn on existing frameworks for quality in qualitative research: including the Standards for Reporting Qualitative Research Framework (SRQR) [15] and specific guidance for quality practice in reflexive thematic analysis [16].

In qualitative research, we are less concerned with reliability and generalisability but attend more to validity, transferability and trustworthiness [17]. To increase the validity of our results, we included multiple researchers in the processes of data collection, coding and analysis; challenging our own assumptions and identifying potential 'blind spots' that any one of us might have had with regards to this topic. We met regularly throughout the course of the research to discuss emerging observations and all members of the research team were actively involved in developing the written report of this study. We also presented our preliminary findings to both mental health colleagues and health and social care workers in a variety of forums to discuss the face validity of our emerging themes. Two healthcare workers who took part in the interviews provided feedback on our analyses as a form of member checking.

In this study, we have not sought to achieve a representative sample of participants with the intention of generalising our results to all health and social care workers in the UK. However, we deliberately tried to include as varied a group of participants as possible in order to explore the diversity of experiences and views of support during the pandemic, to increase the potential transferability of our findings.

To increase the trustworthiness of our interpretations, we have sought to be transparent about the research team conducting the study and the lenses through which we have viewed this data. Below we provide quotes from participants to illustrate and evidence our analyses.

## 3. Results

Twenty-five frontline health and social care workers volunteered and took part in the study. The gender, roles, settings and geographical locations of participants are shown in Table 1.

Interviews took place between June 1st and July 23rd, 2020; and lasted between 30 and 77 minutes. Ethnic origin was not recorded at the point of recruitment, although several participants described themselves as being from minority ethnic groups during the interview.

Experiences and views of psychosocial support during the pandemic were organised into six domains according to potential sources of psychosocial support. These domains are not themes per se but were areas identified by the workers when asked about what sources of

**Table 1. Participant characteristics.**

| Gender | |
|---|---|
| Female | 17 |
| Male | 8 |
| **Role** | |
| Doctor | 7 |
| (Junior) | (4) |
| (Consultant) | (3) |
| Nurse | 9 |
| Healthcare assistant | 2 |
| Physiotherapist | 3 |
| Occupational Therapist (OT) | 1 |
| Paramedic | 1 |
| Care home worker | 1 |
| Mental health care worker | 1 |
| **Setting\*** | |
| Intensive Care Unit (ICU) | 6 |
| Accident & Emergency (A&E) Department | 5 |
| General hospital/COVID wards | 8 |
| Older adult wards | 1 |
| Nightingale Hospital | 1 |
| Ambulance Service | 1 |
| Care home | 4 |
| Psychiatric inpatient setting | 1 |
| **Geographical location by UK region** | |
| London | 9 |
| South East England | 3 |
| South Central England | 2 |
| South West England | 2 |
| Midlands/Central England | 3 |
| North East England | 5 |
| Scotland | 1 |

\*Several participants worked across more than one setting during the peak of the pandemic.

support were important to them. Within these domains, we identified a number of pertinent inductive themes (see Table 2), which are described below and illustrated with quotations.

### 3.1. Self

This category describes how workers sought to support themselves during the pandemic.

**3.1.1 Just getting on with it.** The predominant narrative amongst the frontline workers when they spoke about how they coped during the pandemic was one of "just getting on with it". This mentality was attributed to being a positive person, being part of the job, or simply not having a choice.

> *"yeah, you just get on with it", it's your job."* Ella, ICU nurse

> *"I have to say, it was tough, yes. But there isn't anything I could have done about it really. I just had to get on with it."* Nick, Healthcare assistant

Finding purpose and meaning in the work they were doing helped many of the frontline workers to cope and maintain motivation.

> *"I'd try and draw on the positives that were happening through this, people being discharged, people surviving this against so much past medical history. That was really positive and reassuring to hear".* Katerina, OT

This focus on "just getting on with it", for some however, led to them failing to recognise or subjugating their own needs. Yara, an A&E nurse told us:

**Table 2. Themes (organised within domains of support).**

| |
|---|
| 1. Self |
| 1.1 Just getting on with it |
| 2. Family and friends |
| 2.1 Competing demands |
| 2.2 Separation and sacrifice |
| 2.3 Not in our boat |
| 3. Colleagues, peers and teams |
| 3.1 In the same boat |
| 3.2 Tensions and transitions |
| 3.3 Burden of responsibility |
| 4. Organisational Support |
| 4.1 Practical needs |
| 4.2 Information, communication, and consultation |
| 5. Media and the wider public |
| 5.1 Recognition and awareness |
| 5.2 A double edged sword |
| 5.3 Unhelpful heroic narratives |
| 6. Psychological support services |
| 6.1 Awareness |
| 6.2 Accessibility |
| 6.3 Not for me–reluctance and stigma |
| 6.4 Value of expertise |

"*I cried a lot. Felt rubbish about going to work. But to be honest, I think I just buried it all because I had to.*"

Staff sought opportunities to engage in existing coping strategies where possible, such as exercise or spending time with family, although several acknowledged that they were not able to engage in previously enjoyable activities as much as before, due to working extra hours, being too tired, or not being able to access them due to lockdown and social restrictions. Many workers commented on not having been able to take time off work or use annual leave during the pandemic and several acknowledged increasing alcohol consumption in the absence of other mechanisms to relieve stress.

## 3.2. Family and friends

Participants described how family and friends could be an important source of support, but also engendered feelings of worry and responsibility. They also felt that family and friends were often unable to relate to what they were going through.

**3.2.1. Competing demands.**   Some participants commented on having the opportunity to spend more time together at home with family during lockdown which they valued. For others, trying to manage the competing demands of work and family commitments was a source of stress. Most participants had additional caring responsibilities and talked about juggling childcare with working spouses, not being able to use grandparents for support and often having to take on extra tasks for vulnerable family members who were shielding. Amy, a physiotherapist in ICU told us:

"*Seeing the patients wasn't the emotional bit- we're used to seeing poorly patients- actually having to then deal with your home life and the change to your home life, I think that impacted on it hugely and that was probably more stressful for me than actually working in the acute hospital*"

Similarly, Daniella, a mental health worker described:

"*On my days off I was getting more exhausted because the shopping was taking. . . I was doing shopping for my household, my partner, my mother, his mother and for work.*"

**3.2.2 Separation and sacrifice.**   Many workers expressed concerns about contaminating their families. Participants talked about keeping their families away from other people, and even keeping themselves away from their own families.

Katerina, an OT in London, told us about her fears of contaminating her vulnerable partner who was shielding, leading her to live in a caravan on their driveway for 14 weeks during the peak of the pandemic.

"*It felt like that was the right thing for us to do to try and protect him, because if he was going to get it, it would be only through me. . .and the thought that he could've caught that from me was just overwhelming, so that was the thing that I could do, was to help and keep working, but also protect him at the same time.*"

As a result, several workers talked about feeling isolated, although perceived this as a necessary precaution.

"*I still don't feel very comfortable going to visit my family or any of my friends because I don't want the risk of them having anything and giving it to me and me passing it to the home, or anyone in the home having anything and then me passing it to my family. I feel very–quite a bit, like, isolated from the world, and more isolated than people that aren't working on the frontline, because we have a lot of weight on our shoulders of the places where we have to be.*" Joy, Carer, Dementia Care Home.

**3.2.3 Not in our boat.**   Many participants talked about the importance of keeping in touch with friends and family, often online. Some valued having the opportunity to talk to people outside of health and social care about things that were not work related. Workers valued the support of family and friends, both emotional and practical.

Many participants, however, acknowledged that it was often hard to talk to friends and family who might not be able to relate to what they were going through–who were "not in the same boat".

Olivia, a Physiotherapist told us:

"*Nobody really understands what it was like. I can come home and kind of explain it to my husband, but he doesn't work in a hospital, he doesn't really know what it is like working in intensive care anyway, and so, you couldn't really come home and tell him what it was like, cause nobody ever really can accept what it's like unless you are working there as well.*"

Workers also did not want to burden others. Ester, a Nurse, described trying to talk to her daughter:

"*I tried to talk to her once, but then she said she had her own anxieties and problems with Uni work, and exams and things like that. So, I tried to talk to her once, but she got so upset, that after that I can't put this on her as well.*"

## 3.3. Colleagues, peers and teams

For the most part, participants spoke positively about their experiences of support from colleagues and peers, united through shared experience. However, this was not straightforward, and peers were not always a source of support.

**3.3.1 In the same boat.**   Participants described a strong sense of camaraderie and peer support. In contrast to family and friends, workers felt that colleagues could relate to their experiences.

"*We all support each other, because we're all in the same boat, we're all sort of going through the same emotions.*" Joy, Carer, Dementia Care Home.

Workers valued being able to talk to someone who could understand and relate to what they were going through. When asked who she found it most helpful to talk to, Olivia a Physiotherapist said:

"*definitely the colleagues that I worked with, being able to share that experience with them and talk it through, was really, really helpful. Because, obviously, they really understood how you felt, because 99% of them felt exactly the same way.*"

Workers particularly valued colleagues knowing and understanding the unique demands of their role and would often seek their support over more formal sources.

*"I personally, I would probably rather, I've had a few colleagues that I've used as confidants rather than speaking to somebody outside my circle. I know there are well-being apps and things being rolled out to people free of charge which were suggested to me, but I've not used them, in all honesty."* Michael, Doctor.

Workers appreciated colleagues for being able to normalise and validate emotional responses and appreciated the opportunity to share decision making. They also particularly valued support from colleagues and peers as it was usually easily accessible and timely at the point of need.

**3.3.2 Tensions and transitions.** Relationships between colleagues were not, however, always straightforward or supportive. Whilst many groups were brought closer together by shared experience, others were separated, and divisions were observed between those perceived as doing more and those perceived as doing less. Resentments brewed between those who accepted redeployment and those who were not asked or who refused. As some groups bonded, fractures were amplified between different teams, departments and wards.

Many staff were redeployed into new roles and/or new locations during the peak of the pandemic and this was a particular source of frustration and tension.

*"I got moved to a different team of people to work with that I didn't know and they didn't really know me or my skills or my background or whatever. . . they didn't know what I was able to do and not do, and then I wasn't being used effectively. That was quite frustrating. Or you know, they would change the times they're having meetings and stuff and I wouldn't really be updated. . .I felt like I wasn't being taken care of by their team. Particularly when like, I don't normally do this, I'm already out of my comfort zone a bit here. . .We were kind of the outsiders, and it still is. . .I feel like an outsider."* Alice, Physiotherapist.

Several redeployed staff talked about being *"outside of their comfort zone"* leading them to look for more support from new colleagues, but not always finding it.

*"you go for a break you didn't know anybody, so it was quite lonely on your breaks. So, you kind of just sit there and saw people talking to each other and ranting. But I did not have anybody, because I did not know anybody."* Ester, Nurse.

On returning to previous teams, support was still not always necessarily forthcoming, as Ester went on to tell us.

*"The week I finished my redeployment, I came back to my normal role, and had a bit of a breakdown, because there was nobody here. My colleague went home for annual leave, I couldn't find my managers."*

**3.3.3 Burden of responsibility.** Workers also described a sense of burden in caring for others. Several worried about being a burden when they were working in a new role or environment. Some talked about feeling burdened by colleagues. Others described feeling a burden of responsibility to look after others.

Workers worried about whether they were doing the right thing and giving the right advice to colleagues. They also noted how it was harder to support others when they themselves were feeling depleted.

"*You felt responsible for the nurses that were there on the first shift. On top of everything, you had the added pressure of, you know, you felt responsible making sure that they were okay, when you yourself were not 100%*". Ester, Nurse.

Similarly, Megan, an A&E nurse described how workers did not want to burden colleagues further when everyone was going through the same stresses:

"*We're a really close team but because everybody is in it, there isn't that support around you because everybody needs to be for themselves. It is a time when everybody needs to be quite selfish but that means that there isn't anybody there really asking how you are because everybody is asking themselves how they are.*"

Later, Megan spoke about this as a barrier which prevented workers from seeking peer support from designated colleagues:

"*There are posters in changing rooms and things, saying like 'if you need someone to talk to. . .' and there are the named people who have put themselves forward for if you want to talk to them, but you almost feel like they are in this job as well and I'm going to come to them with something they are doing as well and it's hard, you almost feel like I don't want to bother them with a problem when they are going through it as well.*"

### 3.4. Organisational support

**3.4.1 Practical needs.** Staff valued organisational responses which addressed their basic human needs during the peak of the crisis. This support was appreciated both on a practical level but also in that it conveyed value of the workers themselves. Workers spoke about free meals, access to free parking and areas to rest and recover being important.

"*They provided free parking which I think staff really valued and they also provided free meals. . . and little things like that have really helped because it just makes people feel more appreciated.*" Amy, Physiotherapist.

This kind of practical support was prioritised by many during the peak of the pandemic. As Anna, a Consultant in A&E said:

"*We appreciate the basic things done well. . .making sure that we get our breaks, that we have access to hot food in our department. Simple things like that are actually really important. . .I don't think that putting on meditation or resilience training are necessarily the best way to help and support staff. I think that making them feel like their basic needs are helped and they are valued, are probably the most important things.*"

Whilst this practical support was important, some pointed out inconsistencies in messaging, for example, free food including many sugary and unhealthy snacks. And whilst staff appreciated these resources, they often noted that in reality, having the time during their busy shifts to access them was impracticable. A couple of participants also commented on how resources were positioned, for example, how a rest area being labelled a "wobble room", could perpetuate stigma, as staff did not want to be seen as someone "having a wobble".

Whilst these forms of additional support were appreciated by the workers, when they were subsequently taken away, staff felt de-valued, which was reflected in lower morale.

"*The free snacks and the free coffees go away and everyone's sort of like oh, back to business as usual, people have forgotten that actually, the rest of the time we're all trying to do a really good job and help people, I think that's when everybody's going to need a bit of a morale boost.*" *Alice, physiotherapist*

Workers appreciated flexibility in their working patterns, especially when they had other caring responsibilities or were vulnerable members of staff who needed to shield. Staff valued opportunities to take breaks, although many commented that they had not been able to take time off work and use annual leave.

"*I haven't had a lot of time off. The whole annual leave situation [laughs] not to bag on my trust and my hospital but that was non-existent for a period of time. I haven't had annual leave, I am due to have it so that is fine but I know that more than ever in my life, I am hanging out for a bit of time off because I do feel exceptionally burnt out.*" *Alyssa, Doctor, London*

Workers also shared concerns about patients who were not being treated during the peak of the pandemic and the burden of growing waiting lists. This as well as increasing rates of COVID compounded the sense expressed by many of having had insufficient time to recover.

**3.4.2 Information, communication and consultation.** Workers sought consistent information and clear guidance. They especially wanted clear communication around PPE and safety procedures, but noted that recommendations changed daily, leading to confusion and inconsistencies.

"*The trust was very good in that they had their message that they were filtering down but they didn't filter it down consistently and that was a big level of frustration so it was one day oh, you need to be doing it this way, the next day it was some other way, and everyone interpreted what they meant differently.*" *Amy, Physiotherapist*

Workers talked about problems with communication within and between teams. They noted that many methods of communication were problematic, with staff rarely having time to check emails. Not all staff had equitable access to information, with agency and contractual employees not having access to NHS email accounts, via which information was often disseminated.

Workers wanted communication to be two-way and wanted to be consulted in the process of decision making. Some gave examples of feeling listened to and involved, more spoke about their views not being invited or taken into account.

"*It has been a bit frustrating I think we feel from an A&E point of view that we haven't been that involved. There are these task force that we should we have some representation at, and we haven't been involved. So, I think that we have been expected to respond to things but without being involved in the strategic management.*" *Anna, Consultant, A&E*

At worst, the workers felt like their views were silenced and invalidated by organisational and governmental responses.

"*They should listen to the people on the frontlines, that was the major problem because there was a massive mismatch between what the government kept on saying, what the NHS bosses kept on saying and the people on the front lines saying 'no, we don't have that. We don't have what we need to keep us safe' and the government were trying to silence the people who were*

*working on the front lines. . .they were treating us like we were stupid like making us doubt our memory like 'did this really happen? You're saying we had all the equipment, why is it we don't have it then? Why are we getting ill?'" Alex, Agency Nurse.*

### 3.5. Media and the wider public

Participants described mixed feelings about support offered by the media and the wider public. On the one hand, they appreciated the recognition and value of their work. On the other, they felt such support was often short-lived and many-times unhelpful.

**3.5.1 Recognition and awareness.**   Healthcare workers appreciated media coverage of their work when it raised awareness of important issues in realistic ways. For example, participants from allied health professions, such as physiotherapy and occupational therapy, told us that they valued coverage of their professional roles in healthcare settings which they felt were previously poorly understood by the public.

The media was also valued for its role in raising awareness of the challenges that NHS workers were facing and advocating for resources such as personal protective equipment (PPE).

**"***I think that what the media has done is a good thing because it has said look, we need to have this recognition that the NHS as a whole, it can't handle something like this, and they do struggle, and this is going to be a massive knock back for the NHS, like, they're going to struggle to come back from it a lot. I just feel like the media portrayed it well in saying, there needs to be more help, there needs to be more PPE, stuff like that." Will, Paramedic.*

Participants did, however, also highlight groups of workers who they felt were forgotten by the media and largely absent in their representations, but equally affected by the pandemic, including care home workers, porters, cleaners and mortuary workers. Several participants commented on resentment harboured towards healthcare staff by other keyworkers or furloughed employees who felt overlooked by the media and public.

**3.5.2 A double edged sword.**   Whilst workers appreciated positive media coverage and public gratitude, most expected that this would be short-lived. Participants expected that they would soon be "forgotten" or worse, anticipated that a "backlash" would inevitably follow.

Zahra, a junior doctor told us:

*"I'm worried there's gonna be a backlash when people are seeing that now we have to put all the cancer treatment on hold, and all this stuff, all these really sick people that haven't been coming in, and GPs are gonna be completely on their knees after this, and its already been impossible. So, then the backlash is gonna be big because it's gonna be OUR fault that nobody can be seen."*

Megan, an A&E nurse, captured the double-edged sword of the media portrayal of health care workers:

*"I think that we're the heroes when people want us to be and we're the villains when people want us to be. We are the scapegoat for everything aren't we so, we're the national treasure when we want to show off to the rest of the world but it's not long until we are back to being slated."*

Many workers also went on to talk about the hypocrisy of public and media support. For example, Katerina, an OT, said:

*"The clap for carers, I really liked it at the beginning, I felt it was something really positive that the community felt they could do to show their appreciation. . .but then, as time sort of*

*went on, and we saw more and more people out in the parks, it almost became a little bit of an insult. My team felt like when we would go outside and see people just hanging around in groups, it was just. . .are you the people that are clapping for us and then also going off to do that? It really just seemed contradictory."*

**3.5.3 Unhelpful heroic narratives.**   Several participants commented that media portrayals were inaccurate, tending to glorify healthcare workers' roles and present a positive media story.

"*It just showed the glory of things, you know, people ringing a bell as they left the ward and things like that but it didn't say how many nurses had to move out of their own houses and how many had to cut contact with their children as some people had to. I don't know, I don't think it showed maybe the weight on the shoulders of all of us and yeah, I do feel like the clapping was widely broadcast and showed how lovely everyone was being but I don't think anybody is talking about what it really means to support the NHS and frontline workers*" Megan, A&E Nurse*

Similarly, Michael, and A&E doctor told us:

"*I think most of the things that I've seen have tried to be a bit upbeat and I don't think all of the negatives have come through. I don't think the fact that hospitals were literally like war zones with all of the chaos that was going on has necessarily come out in the media.*"

Heroic narratives were seen as particularly unhelpful and were not welcomed by any of the workers in our sample. Several described this as "over the top" when they perceived themselves to just be doing their jobs; jobs they did all year round, not just during the pandemic. Many participants felt that such narratives detracted from debates about adequate pay and protection.

"*It makes me feel really uncomfortable, with people clapping and all that stuff, when really, you know, what would've been a lot more helpful is PPE and some proper staffing to begin with.*" Zahra, Junior Doctor*

"*Being called a hero or an angel is deemed to be sufficient reward for managing down salaries. . .it is very easy to say you're doing a fab job, you know. . .but it's about how you support your staff in a consistent and meaningful way. So, the notion of heroes and angels, I don't think is one that I have found particularly valuable or helpful. Appreciation is nice, but it's not heroism, it's doing the job that you are paid to do.*" James, Nurse, Scotland.*

For many, this appeared to serve the function of perpetuating a view of health and social care work as a vocation, and a sacrifice that workers were willing to make, akin to military personnel.

"*The Thursday night clap was always very, really like, angering and made us just so frustrated. . .made us feel like "ok, so you think that will be enough for us? Don't give us safe working conditions and hazard pay", but instead, they brought in this idea of heroes, like militarising the whole response, like our death would be seen as a sacrifice rather than absolutely due to inadequate response from the employer, from the government. That made us so angry. The glorification of the NHS worker, instead of giving us what we need which is proper protection.*" Alex, Nurse*

Furthermore, some workers also observed that such narratives devolved responsibility from the wider public.

"*Placing nurses as heroes devolves responsibility from others...we're not heroes, we're all just doing our part...but everybody has their part to play in this.*" Katerina, OT.

### 3.6. Psychological support services

Support from psychological therapy services, when available, was largely valued, and those who had accessed them, or knew others who had, spoke positively about them. However, there appeared to be large disparities in what was available and significant barriers to access.

**3.6.1 Awareness.** There was a striking variety of experiences amongst the workers in our sample. Some felt psychological support services were well advertised and available, although notably not many of these participants felt that they themselves were in need of them. Many others were unaware of what psychological support was available.

Some participants highlighted a lack of recognition of mental health issues in the first place.

"*I think firstly, there needs to be a way of recognising it because everyone has been so involved that it has been hard to recognise that people are falling apart, one of the people went off with an acute psychotic episode and actually in retrospect we should have picked that up earlier and one of my other colleagues has gone off with very bad depression in the last week and the signs were there but it's a matter of how you try and spot that.*" Michael, Consultant.

A few participants who had received wellbeing or Psychological PPE training early on in the pandemic valued this; some others thought this desirable, but for most, this was not something they had experienced.

Communication about what psychological support services were available was often unclear and inconsistent, leaving staff uncertain about what support was available and confused about how to access it.

"*There hasn't been psychological support available. Or at least I didn't have time to look for it. And if it was there, I didn't see it. It wasn't made available, it wasn't visual...I certainly haven't seen anything.*" Grace, ICU Nurse

"*Lip service is given to numbers you can call and things like that. But I wouldn't know how to access it. I don't know their numbers; I wouldn't know who to contact.*" Jasser, Consultant

**3.6.2 Accessibility.** Even when staff were aware of support being available, they often seemed to feel that this was quite remote, irregular and not easily accessible. When asked what mental health support was available, Ester, a redeployed nurse, told us "*they just said that the wellbeing team is available and to contact them if we had any need of it*". Anna, a Consultant in A&E also described:

"*We didn't have regular access to a psychologist in the department. They were kind of infrequently coming and touching base with us. I don't know how great the uptake would be if they had come down regularly...I was hoping we would be able to have some support on a regular basis.*"

When staff did try to seek help, they were not always able to access it. Yara, an A&E nurse told us:

*"We were told if we wanted to speak to someone, we could phone occupational health. But every time you tried to phone occupational health it was engaged."*

Similarly, Anna, Consultant in A&E, talked about problems accessing counselling services in her hospital:

*"People have tried to take up these interventions, but they haven't managed to actually get through to anybody. It's just that the phone lines are busy, or they have been unlucky in terms of when they are trying to call. . .So, although technically the provision is there, I haven't actually spoken to anyone who has managed to speak to one of the counsellors."*

There were also many practical and structural barriers to accessing psychological support services. Workers commented on services usually only being available in working hours, Monday to Friday, which did not correspond with the shift patterns that most staff worked during the pandemic. Many staff commented that they were too busy during their shifts to be able to take time out for such appointments, and understandably did not want to attend after the end of long shifts, or to come in to work on their time off.

*"There were a couple of things set up like one was to drop in and have a cup of tea and a chat about how people are feeling but I don't think people ever realised like we are not going to be able to just pop in and have a cup of tea in the middle of the day. They were good ideas and people meant well but in reality, nobody wants to stay at the end of a shift to talk. . .nobody is going to get to talking because there just isn't time when people are at work." Megan, A&E Nurse*

There also appeared to be striking inconsistencies in the provision of mental health support across services, as noted by several workers when they moved between locations and specialties. There were particular barriers to access for staff who were not employed by the NHS, restricting access to many NHS based services for social care staff and agency staff not on NHS contracts.

Redeployed staff also found it especially hard to access services when transitioning between services and line managers, often key points of access to support services.

*"I felt isolated and left on my own. . .and then my manager said: 'Once you're redeployed, I'm not your manager anymore. It's the COVID-19 centre that manages you.' It's very difficult to navigate around the system and find who actually is managing you." Ester, redeployed nurse.*

**3.6.3 Not for me–reluctance and stigma.**   Most of the staff we interviewed had not accessed any psychological support services themselves. For several participants, this was because they perceived that they did not need this, or preferred to seek support from colleagues, friends and families. However, many of the participants in this study did describe feeling very distressed by their experiences of working during the pandemic. Strikingly, several commented that the interview was the first time they had had the opportunity to talk about and reflect on their experiences and they found this helpful. Some said that more mental health support would have been helpful for their teams but tended to dismiss this for themselves. This highlights a potential gap between need and engagement.

Some staff groups seemed to hold the perception that such services were not intended for people like them. For example, some doctors and allied health professionals perceived the

services as only applicable to nurses. Others pointed to more explicit stigma as a barrier to seeking help. Joseph, A&E Healthcare Assistant said:

"*I think there's a huge stigma with mental health. Maybe not as much as it used to be. But I think there's definitely still a stigma. Probably more so for men. . .People feel that they can go but it might affect how people perceive them as a person. How they're perceived as being a strong individual or something like that.*"

Similarly, Nathan, a Junior Doctor told us:

"*The problem with healthcare is that mental health is slightly stigmatised in healthcare workers and people don't want to admit that there is a problem. . .they stress a culture of resilience and I don't think anyone wants to be seen as being unable to cope with anything.*"

**3.6.4 Value of expertise.** Whilst many of the workers in our sample sought support from colleagues and peers who could relate directly to their experiences, others, at other times, valued the expertise, neutrality and confidentiality of being able to speak to a mental health specialist.

Several participants commented that informal and reflective team sessions facilitated by a mental health professional had been beneficial. Workers worried about saying the right thing or giving appropriate advice, so seemed reassured when sessions were facilitated by someone with psychological expertise. In a traditionally hierarchical health system, some participants specifically desired advice from the experts most qualified to give it.

"*My view. . .is that it [psychological support] should be a clinical psychologist. Because that's just, you know, I do the type of surgery I do because that's what I do, and no one else could do it. And you guys do what you do. So, I think it should be a consultant clinical psychologist leading a service.*" Jasser, Consultant.

Workers valued dedicated time to talk and be listened to, which they did not think they could expect from colleagues and peers. The also particularly valued a confidential and non-judgement space in which to talk, and several commented that they did not think it always appropriate to speak to supervisors and line managers about how they were feeling.

"*I think it's important that the other person isn't directly associated with your job and then you can just have the opportunity to express really openly without anything getting back to your team or you bosses about how you really feel.*" Nathan, Junior Doctor.

For most, accessing psychological support services in their own hospital was most desirable and convenient. However, there were a couple of notable exceptions, where mistrust in their organisation and management had been fostered, some workers preferred to be able to access a service that was outside of their immediate organisation.

Several workers also talked about access to individual sessions being important, in addition to group and team support, as they did not always feel able to talk openly in front of colleagues or had experienced certain individuals in groups dominating.

"*I just think it's difficult to express how you really feel, if you are with peers you might not want to open up or display any kind of sign of weakness.*" Nathan, junior doctor

Several participants in this study spoke about the benefits of formal mental health support being appreciated after the peak of the crisis. Workers recognised that during the peak of the pandemic they tended to focus on the work at hand and be *"running on adrenaline"*. After the peak of the crisis had past, they acknowledged that they had more time to reflect, and for many, this was when they recognised the greatest impact on their mental health.

*"Now is the time that people are developing some of the problems. At the time when everybody was immersed and working, there were a few people developing problems, but actually I think that the PTSD and the aftereffects are coming now." Michael, Consultant.*

Several participants also commented on a longer-standing lack of psychological support, and that such support should be made available in the longer-term, not just during the crisis.

## 4. Discussion

In this study we sought to better understand frontline health and social care workers' experiences and views about psychosocial support during the COVID-19 pandemic. We have deliberately considered the full breadth of potential sources of psychosocial support in this paper, including individual, team, organisational and societal aspects, as these were raised by workers when asked about what sources of support were important to them.

The results of this qualitative thematic analysis found that workers' experiences and views about support were complex and nuanced. Workers for the most part tended to adopt a "just get on with it" attitude which often led to them not recognising or subjugating their own psychological wellbeing. They valued emotional and practical support from family and friends but often worried about them and felt they could not relate to their experiences as they were "not in the same boat". They valued the shared experience with colleagues who were "in the same boat" and often looked to peers as their first line of support. However, peer relationships could also be complicated, and many staff did not want to burden, or be burdened by, colleagues. Workers were ambivalent about support shown by organisations, media and the public. Whilst they valued psychological support services, there were many disparities in provision and barriers to access. These findings have important clinical and research implications for better supporting the psychological wellbeing and mental health of the frontline health and social care workforce.

### 4.1. Peer and family support

Colleagues and peers were most workers' first line of support. They valued talking to peers in the first instance as they could relate to each other's experiences and support was usually easily and immediately accessible. However, there were several caveats to this, and peer support was certainly not straightforward.

Peer support was sometimes experienced by workers as a burden. They worried about burdening, and being burdened by, colleagues, who were by definition going through the same stressors as they were. The mutuality of shared experience could bring workers together, but empathy for others' distress has also been well established as a risk factor for vicarious traumatisation [18]. Therefore, whilst peer support may be an important component of psychosocial support for health and social care workers, [19] the results of this study show that it is likely best embedded in a wider system of support, including peers, organisations and mental health professionals, so as not to place too much additional pressure on peers, at a time when their emotional resources may already be depleted.

The results of this study also showed that whilst certain groups bonded through shared experience, other teams and services experienced fractures and conflict. Social psychology has made us aware that the strong formation of an "in-group" necessitates greater distancing and rejection of the "out-group" [20]. This was particularly apparent during COVID-19 with conflicts between wards, services and localities. This made it particularly difficult for staff who were redeployed during the pandemic, who did not always feel included in their new ingroup but were no longer part of their old ingroup and could be left without clear lines of management support. Other quantitative research [3] has shown that being redeployed was a significant risk factor for health and social care workers developing PTSD in the context of the pandemic. Additional psychosocial support for redeployed staff is therefore likely to be of paramount importance.

The practical and emotional support of family and friends was valued by health and social care workers but could also engender feelings of worry and responsibility, and for many, family life brought with it additional competing demands for childcare and care-giving. Workers often believed that family and friends could not relate to what they were going through and sought not to burden them. The experiences of family members of workers in high-risk occupational roles have received little research outside of a military context [21] and would warrant further research in order to better understand how to support the families and friends of health and social care workers.

## 4.2. Mental health awareness

Staff often did not attend to the state of their own, and their colleagues', mental health, indicative of a lack of awareness of mental health issues in some physical healthcare settings. Mental health awareness training could facilitate better recognition and prioritisation of their own and others' psychological wellbeing and has a demonstrable impact on improving healthcare worker wellbeing and burnout [22].

Greenberg and Tracy [23] have advocated for supervisors to be trained in having "psychologically savvy conversations". This could certainly go some way to mitigating the lack of awareness of mental health issues in the health and social care workforce. However, the workers in this sample talked rarely about supervisors and managers as sources of support. Several also talked about deliberately wanting to access support that was outside of their line management structure. We cannot assume that this means supervisors are not important, but nor does it mean that we can assume that they are. It may therefore be equally, or even more important, for mental health awareness training to be made available to all staff.

Furthermore, awareness of mental health issues alone is unlikely to lead to behaviour change and several participants in this study spoke about enduring stigma associated with transparency around mental health difficulties. Stigma about mental illness amongst healthcare professionals has previously been well-documented and noted as a workplace culture issue and a barrier to help-seeking [24]. It is likely that top-down encouragement, role modelling by senior staff and cultural change will be necessary to increase psychological safety and subsequently workers' willingness to talk about mental health in the health and social care workplace.

## 4.3. Organisational, media and public behaviour

Workers appreciated public support and felt valued by gestures of media and organisational support. However, heroic narratives were unhelpful and were in contradiction to the professionalism of health and social care roles, detracted from debates about pay and protection, and

at worst, created a barrier to staff seeking support. After all, heroes do not struggle; angels do not get PTSD.

Workers desired clear and consistent information from government and organisations, and this appeared to mitigate anxiety and stress. They also wanted to be consulted and collaborated with more, but few had experienced this. Consultation and co-production with frontline staff is going to be essential in establishing systems of support which are likely to be most effective, acceptable, and sustainable.

Workers felt gestures of practical support conveyed value of the workers themselves. However, when resources were taken away, this had a deleterious effect on mental health and left workers feeling de-valued and demoralised. An example of this which has previously been seen was when doctors' messes; hubs for rest, refreshments and social interaction amongst physicians, were discontinued, leading to demoralisation and disaffection amongst medical staff who felt that one of the final vestiges of their wellbeing had been dismissed [25].

Several participants in this study acknowledged that staff morale after the peak of the pandemic was very low. This is consistent with the findings of a recently published Royal College of Nursing survey [26] of 42,000 nurses, which found that pay and feeling under-valued were nurses' primary concerns. Subsequently, 35% of the sample stated they were considering leaving the profession.

Significant steps need to be taken to improve the psychological wellbeing and morale of the UK health and social care workforce and to ensure that the services they deliver to the UK population are sustainable, during the COVID-19 pandemic and beyond. Resources for support need to be made consistently available, and easily accessible to all staff. However, systemic and cultural barriers to access need to be addressed to ensure that accessing such resources is not inadvertently stigmatising. Access to resources also needs to be equitable, within different teams and localities and across the health and social care workforce.

## 4.4. Psychological support services

Workers valued psychological expertise and found sessions facilitated by mental health experts reassuring, as they were often worried about saying the wrong thing. They valued being able to access a confidential and independent space, with less burden placed on health and social care peers. Any effective and acceptable mental health support for frontline workers is therefore likely to need to be flexible and accommodate different preferences, including group and individual support, peer and expert led interventions and in-house and external support services.

The results of this study lend support to emerging evidence that psychosocial support may be most acceptable and effective when delivered in a phase-based way [10], although this warrants evaluation. Mental health awareness training, health promotion and ill-health prevention are likely to be most helpful in advance of crises with practical resources, peer support and informal psychological support most valued at the peak. Formal psychological support is likely to be vital in the early recovery and subsequent period, requiring more assertive follow up of workers, early detection of mental health problems and signposting to evidence-based treatment. Effective screen and treat programmes may be required to identify staff in greatest need.

There also needs to be equitable access to support services for all staff, as many staff groups are not employed directly by the NHS so have fallen outside of support being offered by it. This risks perpetuating systemic injustice in support provision, as those from ethnic minority groups, single parents and those on lower incomes are likely to be over-represented in agency and contractual work [27, 28].

There were notable disparities in provision and barriers to access to psychological support services. Staff were often uncertain about what psychological support was available and how to

access it. When support was available, it was often not easy to access or unavailable at convenient times. This highlights that psychological support needs to be made available within working hours and time protected to attend. To not do this communicates to staff that their mental health is not a work priority.

The findings of this study also suggest that communication about psychological support needs to be more coherent and consistent. Mental health practitioners are likely to have to more assertively engage health and social care staff to make them more aware of services available and overcome stigma in attending. Research with mental health professionals who were surged to provide psychological support for frontline health and social care workers during the pandemic [29] showed that much of the mental health care workforce had received no prior training in the needs of health and social care staff and there has been little research into what psychological interventions are most effective in supporting them. This highlights the need for further training and CPD for mental health care providers, as well as the need to develop an evidence base of effective psychological interventions specifically tailored to the unique needs of the health and social care workforce.

Since the initial peak of the COVID-19 pandemic in the UK, NHS England and NHS Improvement announced an investment of £15 million to fund rapid mental health assessment and treatment for NHS staff [30], including piloting of a number of specialist 'Wellbeing Hubs'. This builds on the commitment outlined in the NHS People Plan 2020–21 published in July 2020, to provide a more supportive working environment for staff in the NHS [31]. However, in the early stages of delivery, many psychological wellbeing services have focused on offering training in "psychological resilience" to individuals. Whilst such primary prevention measures have their virtues, we need to be cautious that such messages do not inadvertently perpetuate stigma about mental ill-health and being seen as "not resilient". We also need to make sure that we do not place more burden on individual health and social care workers to "be more resilient", but rather need to address the culture, systems and structures identified in this study which present very real barriers to their wellbeing.

Finally, all psychological support services will need to be adequately resourced and financed, not just in the short term but also in the long run, in order to protect the mental health of the health and social care workforce, and in so doing, maintain the health and wellbeing of the UK population.

## 4.5. Clinical implications

- There is a need to increase mental health awareness for all staff in health and social care settings

- Top-down encouragement, role modelling by senior staff and culture change are needed to increase willingness to talk about mental health in the health and social care workplace

- Staff who are redeployed or who transition between teams and services are likely to need additional support

- A combination of peer, organisational and professional support, which accommodates flexibility and personal preference is likely to be most acceptable and effective

- Systems of support need to be coherent, consistently communicated and easily accessible

- Staff need protected time during working hours to access wellbeing resources and psychological support services

- More assertive outreach is likely to be needed to engage staff with psychological support and identify those most in need

- Structural and systemic barriers to accessing support need to be addressed, not just individual resilience

- Equity of access to support needs to be ensured between different teams, services and localities, as well as across the health and social care sector

- Psychological support services need to be adequately and sustainably resourced and funded, and not just made available during crises

More collaboration, consultation and co-production of support services, and their evaluation, with health and social care staff is needed

### 4.6. Strengths and limitations

The results of this study should necessarily be considered within the context of its strength and limitations. The analysis of this qualitative data was rigorous, with all steps taken to maximise the validity and trustworthiness of the findings. We deliberately sought to include a varied group of participants from across the UK, including diverse professional groups at different career stages, in order to explore the diversity of experiences and views of support during the pandemic and increase the potential transferability of our findings. Our research team was diverse with considerable clinical NHS experience. Health and social care workers' perspectives were included in the design, delivery, analysis and write up of this paper.

Nevertheless, this study has a number of limitations. Despite our best efforts to include them, we were only able to recruit one non-medical care home worker and no participants from outside of a direct healthcare role, such as administrators, cleaners or porters. More assertive outreach is needed to engage these groups in future research. We also did not record the ethnic origins or ages of our participants. Whilst several workers identified themselves in their interviews as from black and minority ethnic groups, and were from a variety of career stages, this does limit the potential diversity of our sample. Only one participant from Scotland took part and none from Wales. Workers in these settings might have had different experiences in these different healthcare systems. Further research is needed to extend these questions to these groups and allow their voices to be adequately heard. The roles that health and social care workers undertook during the pandemic were multifarious and therefore have not been categorised quantitatively in this study. This study took place over seven weeks in the early post-peak phase of the first wave of the pandemic in the UK. It is likely that workers' needs change over time. Therefore, there needs to be ongoing dialogue with health and social care workers about their needs and preferences.

### 5. Conclusions

This study provides an in-depth analysis of frontline health and social care workers' views, which elucidates many complexities about their relationship with different domains of psychosocial support which were hitherto poorly understood. The results of this study show that a "one-size fits all" approach to providing support is unlikely to be helpful, and that rather a systematic approach to support including peers, organisations and professional support is warranted. Nevertheless, these systems of support need to be coherent, consistently communicated and easily accessible. More research is needed to fully unpack the structural, systemic and individual barriers to accessing psychosocial support. The views of workers from minority professional and ethnic groups need to be assertively included in future research.

More collaboration, consultation and co-production of support services and their evaluation is warranted.

## Supporting information

**S1 Data. Interview guide.** This is the interview schedule which was used to guide the semi-structured interviews.
(DOCX)

## Acknowledgments

We would like to thank all the frontline health and social care workers who gave up their time to take part in this research. We would also like to thank our Expert Reference Group and our health and social care colleagues who provided invaluable guidance on the design, delivery and analysis of this study.

## Author Contributions

**Conceptualization:** Jo Billings, Michael Bloomfield, Talya Greene.

**Formal analysis:** Jo Billings, Nada Abou Seif, Siobhan Hegarty, Tamara Ondruskova, Emilia Soulios.

**Investigation:** Jo Billings, Nada Abou Seif, Siobhan Hegarty, Tamara Ondruskova, Emilia Soulios.

**Methodology:** Jo Billings.

**Project administration:** Jo Billings, Nada Abou Seif, Siobhan Hegarty, Tamara Ondruskova, Emilia Soulios.

**Supervision:** Jo Billings.

**Validation:** Jo Billings, Nada Abou Seif, Siobhan Hegarty, Tamara Ondruskova, Emilia Soulios, Michael Bloomfield, Talya Greene.

**Visualization:** Jo Billings.

**Writing – original draft:** Jo Billings.

**Writing – review & editing:** Nada Abou Seif, Siobhan Hegarty, Tamara Ondruskova, Emilia Soulios, Michael Bloomfield, Talya Greene.

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
