## [Decision Letter · Decision Letter 0]

9 Jun 2021

PONE-D-21-07944

What support do frontline workers want? A qualitative study of health and social care workers’ experiences and views of psychosocial support during the COVID-19 pandemic

PLOS ONE

Dear Dr. Billings,

Thank you for submitting your manuscript to PLOS ONE. After careful consideration, we feel that it has merit but does not fully meet PLOS ONE’s publication criteria as it currently stands. Therefore, we invite you to submit a revised version of the manuscript that addresses the points raised during the review process.

We look forward to receiving your revised manuscript.

Kind regards,

Vincenzo De Luca

Academic Editor

PLOS ONE

Journal Requirements:

2. In your Methods section, please clarify whether pseudonyms were used.

3.We note that you have indicated that data from this study are available upon request. PLOS only allows data to be available upon request if there are legal or ethical restrictions on sharing data publicly. For information on unacceptable data access restrictions, please see http://journals.plos.org/plosone/s/data-availability#loc-unacceptable-data-access-restrictions.

4.Thank you for stating the following in the Acknowledgments Section of your manuscript:

"This research received no specific grant from any funding agency, commercial or not-forprofit

sectors. Dr Bloomfield is funded by a UCL Excellence Fellowship and supported by

the National Institute for Health Research University College London Hospitals Biomedical

Research Centre."

 "The authors received no specific funding for this work."

Reviewers' comments:

Reviewer's Responses to Questions

**Comments to the Author**

1. Is the manuscript technically sound, and do the data support the conclusions?

Reviewer #1: Yes

Reviewer #2: Partly

2. Has the statistical analysis been performed appropriately and rigorously? 

Reviewer #1: Yes

Reviewer #2: N/A

3. Have the authors made all data underlying the findings in their manuscript fully available?

Reviewer #1: Yes

Reviewer #2: Yes

4. Is the manuscript presented in an intelligible fashion and written in standard English?

Reviewer #1: Yes

Reviewer #2: Yes

5. Review Comments to the Author

Reviewer #1: Overall well written and informative manuscript. However the following comments with respect to different sections of the manuscript to be considered:

Introduction: Mentioning stats about mental health burden among UK based health professionals may be added.

Methods: There is no clear description about how sample size was achieved.

Discussion: Mental health awareness and peer support components are two different domains so these should be separately written. Familial support discussion may be added to peer support domain. Add more evidence into discussion part.

Reviewer #2: The current article analyzes UK frontline health and social care workers’ own experiences and views of psychosocial support during the pandemic by COVID 19 by means of a qualitative methodology. Twenty-five frontline health and social care workers were recruited and interviewed remotely following a semi-structured interview guide. Authors showed that workers’ experiences and views about psychosocial support were complex. Peer support was many workers’ first line of support but could also be experienced as a burden. Workers were ambivalent about support shown by organizations, media and the public. The results of this study show that frontline health and social care workers are likely to need a flexible system of support including peer, organizational and professional support.

The paper is well written, theoretically informed and addresses a subject that had not yet fully covered. This study contributes meaningful and relevant knowledge not only to health and social care workers but also to the entire society about the transverse impact of a pandemic.

However, there are some aspects that need to be improved by the authors:

1. The sample description requires more detail (in the manuscript and table 1). For example, ages, job or position, outbreak roles, etc.

2. Inclusion/exclusion criteria must be specified as well as data saturation criteria, and also a description of participants who were approached but refused to participate in the study.

3. Regarding the results, themes/sub-themes need to be revisited (they are mostly descriptive). As a qualitative study, a higher analytical depth is expected.

6. PLOS authors have the option to publish the peer review history of their article (what does this mean?). If published, this will include your full peer review and any attached files.

Reviewer #1: No

Reviewer #2: No

---

## [Author Response · Author response to Decision Letter 0]

12 Jul 2021

We have responded to the editor's and reviewers' comments in a point-by-point response in our uploaded letter.

---

## [Decision Letter · Decision Letter 1]

9 Aug 2021

What support do frontline workers want? A qualitative study of health and social care workers’ experiences and views of psychosocial support during the COVID-19 pandemic

PONE-D-21-07944R1

Dear Dr. Billings,

We’re pleased to inform you that your manuscript has been judged scientifically suitable for publication and will be formally accepted for publication once it meets all outstanding technical requirements.

Kind regards,

Vincenzo De Luca

Academic Editor

PLOS ONE

Additional Editor Comments (optional):

Reviewers' comments:

Reviewer's Responses to Questions

**Comments to the Author**

1. If the authors have adequately addressed your comments raised in a previous round of review and you feel that this manuscript is now acceptable for publication, you may indicate that here to bypass the “Comments to the Author” section, enter your conflict of interest statement in the “Confidential to Editor” section, and submit your "Accept" recommendation.

Reviewer #1: All comments have been addressed

Reviewer #2: All comments have been addressed

2. Is the manuscript technically sound, and do the data support the conclusions?

Reviewer #1: Yes

Reviewer #2: Yes

3. Has the statistical analysis been performed appropriately and rigorously? 

Reviewer #1: Yes

Reviewer #2: N/A

4. Have the authors made all data underlying the findings in their manuscript fully available?

Reviewer #1: Yes

Reviewer #2: Yes

5. Is the manuscript presented in an intelligible fashion and written in standard English?

Reviewer #1: Yes

Reviewer #2: Yes

6. Review Comments to the Author

Reviewer #1: (No Response)

Reviewer #2: I have reviewed the revised version of the manuscript. I realized that all of the recommendations have been considered and addressed adequately. I believe that this manuscript will contribute to the existing literature.

7. PLOS authors have the option to publish the peer review history of their article (what does this mean?). If published, this will include your full peer review and any attached files.

Reviewer #1: No

Reviewer #2: No

---

## [Editor Report · Acceptance letter]

25 Aug 2021

PONE-D-21-07944R1 

What support do frontline workers want? A qualitative study of health and social care workers’ experiences and views of psychosocial support during the COVID-19 pandemic. 

Dear Dr. Billings:

I'm pleased to inform you that your manuscript has been deemed suitable for publication in PLOS ONE. Congratulations! Your manuscript is now with our production department. 

Kind regards, 

on behalf of

Dr. Vincenzo De Luca 

Academic Editor

PLOS ONE